# *Plasmodium falciparum* Development from Gametocyte to Oocyst: Insight from Functional Studies

**DOI:** 10.3390/microorganisms11081966

**Published:** 2023-07-31

**Authors:** Dinkorma T. Ouologuem, Antoine Dara, Aminatou Kone, Amed Ouattara, Abdoulaye A. Djimde

**Affiliations:** 1Malaria Research and Training Center, Faculty of Pharmacy, Faculty of Medicine and Dentistry, University of Sciences, Techniques, and Technologies of Bamako, Bamako 1805, Mali; 2Malaria Research Program, Center for Vaccine Development and Global Health, University of Maryland School of Medicine, Baltimore, MD 21201, USA

**Keywords:** *Plasmodium*, gametocytes, gamete, differentiation, gene expression regulation

## Abstract

Malaria elimination may never succeed without the implementation of transmission-blocking strategies. The transmission of *Plasmodium* spp. parasites from the human host to the mosquito vector depends on circulating gametocytes in the peripheral blood of the vertebrate host. Once ingested by the mosquito during blood meals, these sexual forms undergo a series of radical morphological and metabolic changes to survive and progress from the gut to the salivary glands, where they will be waiting to be injected into the vertebrate host. The design of effective transmission-blocking strategies requires a thorough understanding of all the mechanisms that drive the development of gametocytes, gametes, sexual reproduction, and subsequent differentiation within the mosquito. The drastic changes in *Plasmodium falciparum* shape and function throughout its life cycle rely on the tight regulation of stage-specific gene expression. This review outlines the mechanisms involved in *Plasmodium falciparum* sexual stage development in both the human and mosquito vector, and zygote to oocyst differentiation. Functional studies unravel mechanisms employed by *P. falciparum* to orchestrate the expression of stage-specific functional products required to succeed in its complex life cycle, thus providing us with potential targets for developing new therapeutics. These mechanisms are based on studies conducted with various *Plasmodium* species, including predominantly *P. falciparum* and the rodent malaria parasites *P. berghei*. However, the great potential of epigenetics, genomics, transcriptomics, proteomics, and functional genetic studies to improve the understanding of malaria as a disease remains partly untapped because of limitations in studies using human malaria parasites and field isolates.

## 1. Introduction

Malaria is a mosquito-borne infection caused by an apicomplexan parasite of the genus *Plasmodium* and is transmitted through the bite of a mosquito vector of the genus *Anopheles*. In humans, malaria is caused by six *Plasmodium* species, including *Plasmodium falciparum*, *Plasmodium vivax*, *Plasmodium malariae*, *Plasmodium ovale curtisi*, *Plasmodium ovale wallikeri,* and the zoonotic parasite *Plasmodium knowlesi*. Of the human malaria parasites, *P. falciparum* causes the most significant morbidity and mortality. In 2021, clinical malaria affected 247,000 million individuals worldwide, leading to 619,000 deaths, with the highest burden observed in sub-Saharan Africa [1]. Over the last two decades, artemisinin-based combination therapies (ACTs), intermittent preventive treatments (IPTs), and vector control approaches have significantly reduced the burden of malaria in some areas [1]. However, despite tremendous efforts to control malaria, the emergence of resistant parasites to current antimalarials and the resistance of mosquitos to insecticides are significant obstacles to the global malaria eradication/elimination program [1].

With the renewed interest in malaria elimination and eradication, it has been recognized that intervention strategies should target all stages of the parasite life cycle with a particular interest in preventing transmission [2]. *Plasmodium* transmission from the human host to the mosquito vector and the parasite stages within the mosquito are promising targets for successfully interrupting the parasite life cycle and reducing the malaria burden in endemic areas [3].

The life cycle of *Plasmodium falciparum* is very complex. It includes different hosts, different cell types within the same host, asexual multiplication phases, and a sexual replication phase (Figure 1). In humans, the cycle begins with the inoculation of the sporozoite forms of the parasite by an infected mosquito during its blood meal. These sporozoites migrate through the bloodstream, reach the liver, and enter the hepatocytes, where they multiply asexually. After six to ten days, the rupture of infected hepatocytes releases thousands of merozoites, a new form of the parasite, into the bloodstream. The merozoites invade the red blood cells (RBCs) to initiate the intraerythrocytic cycle. After two days, new merozoites are released, and these offspring enter new RBCs to perpetuate the intraerythrocytic cycle responsible for the clinical manifestations of malaria in humans. A small percentage of merozoites differentiate into sexual forms called gametocytes, which mediate transmission. Gametocytes are morphologically (shape, size, and flexibility) [4,5,6] and functionally different from asexual parasites [6,7,8,9]. Mosquitos ingest male and female gametocytes during a blood meal, which results in a wave of differentiation processes and developmental stages within the mosquito [6,10,11,12]. In the mosquito’s gut lumen, gametocytes transform into male and female gametes. These gametes then fuse to produce a zygote that differentiates into a highly motile and invasive ookinete [6,11,13]. The ookinete invades the mosquito epithelial cell wall to form an oocyst, which in turn will ultimately produce thousands of sporozoites through a process known as sporogony [6,12,14]. The sporozoites will migrate and reside in the mosquito salivary gland [12], waiting to be transmitted to the vertebrate host during a blood meal and perpetuating the parasite’s life cycle.

The drastic changes in both shape and function of the *Plasmodium* parasite to adapt to distinct environmental niches throughout its life cycle rely on tight programming of stage-specific gene expression, as shown by several functional genomics, transcriptomics [15,16,17,18,19,20], and proteomic analysis [21,22]. Understanding the mechanisms that drive the development of gametocytes, gametogenesis, sexual reproduction, and zygote differentiation within the mosquito will likely reveal new avenues to interrupt the parasite life cycle, thus interrupting malaria transmission. Topics of interest for developing transmission-blocking strategies include factors that trigger asexual parasites to commit to sexual differentiation, the molecular players involved in gametocytogenesis, and gametocyte maturation. A better understanding of the mechanisms involved in gamete formation and fusion, the requirements for establishing a successful sexual stage within the mosquito, and the molecular players controlling zygote to oocyst production are also of interest.

Genomics, transcriptomics, and proteomics provide valuable insight into *Plasmodium* biology, including parasite differentiation within humans and the mosquito vector [18,21,23,24,25]. Since the first publication of the *Plasmodium falciparum* reference genome in 2002, a significant number of *Plasmodium* spp. Whole-genome sequences have been released, allowing in-depth comparative analysis [26,27]. These analyses revealed genes potentially linked to phenotypic differences [28,29]. Advances in sequencing technologies and analytical techniques allow a more detailed exploration of the parasite genomes, improving our understanding of the parasite ecology and epidemiology [30,31,32]. Transcriptomics and proteomics improved our knowledge of RNA and protein expression dynamics and their regulation throughout the parasite life cycle [33,34,35].

The cycle begins with the inoculation of the sporozoite forms of the parasite by an infected mosquito during its blood meal. These sporozoites migrate through the bloodstream and reach the liver. The sporozoites will invade the hepatocytes and initiate the exoerythrocytic developmental cycle. The infected hepatocytes release thousands of merozoites 6–10 days after the initial invasion. This new form of the parasite will enter the bloodstream to invade the red blood cells (RBCs) and initiate the intraerythrocytic cycle. Within the intraerythrocytic developmental cycle, a small percentage of the parasite commits to sexual differentiation, resulting in the development of the gametocyte forms. The gametocytes will be picked up by the mosquito during their blood meal. Within the mosquito, the sporogonic cycle will take place.

Gametocytogenesis and *Plasmodium* development in the mosquito have been the subject of several comprehensive reviews [36,37,38,39]. However, these reviews often focus either on gametocytogenesis in the human host or parasite development within the mosquito vector. In this review, we aim to cover the *Plasmodium falciparum* developmental process from the pre-sexual stage in the human host to the ookinete stage of the mosquito vector. The review is based on studies conducted with various *Plasmodium* species, including predominantly *P. falciparum* and the rodent malaria parasites *P. berghei*. Although *Plasmodium berghei* is a rodent malaria parasite, the availability of fast and efficient experimental genetics techniques to access the complete in vivo life cycle made rodent malaria parasites valuable models for understanding some aspects of *P. falciparum* biology. The review highlights studies assessing the mechanisms involved in the parasite-stage-specific gene expression, sexual stage development in both the vertebrate and mosquito vector, and zygote to oocyst differentiation. This review starts with an outline of genome organization, general gene expression, and regulation mechanisms in the intraerythrocytic asexual stage. Then, we describe the mechanisms driving the development of gametocytes, gametes, fertilization, and zygote to oocyst differentiation. Finally, the review highlights gaps and potential targets for developing new classes of antimalarial drugs or vaccines.

## 2. *Plasmodium* Genome Organization and General Mechanism for Gene Expression Regulation

### 2.1. Plasmodium Genome Organization

The *Plasmodium* parasite genome of ~18–30 Mb is organized into 14 chromosomes containing about 5300 protein-coding genes, with many sub-telomeric multigene families [26]. The packaging and organization of genomic DNA into chromatin constitute an essential regulatory mechanism for *Plasmodium* gene expression [30,32]. As in model organisms, specific nucleosome positions restrict the accessibility of regulatory DNA elements and thus can be predictive of gene transcription state [40,41]. The *P. falciparum* nucleosome organization consists of approximately 155 DNA base pairs (bp) around a histone octamer [42]. Site-specific analysis of nucleosome positioning has revealed that nucleosomes are best positioned when flanked by AT/TA dinucleotides [41]. *Plasmodium* histone molecules are characterized by distinct biochemical properties thought to modulate both nucleosome stability and their rapid displacement in a temperature-dependent manner [41]. The downstream regions of transcription start sites are unusually depleted of nucleosomes, presumably to facilitate the recruitment of the basal transcription machinery [43].

A comparative analysis of chromosome organization in different plasmodial species revealed that the two most pathogenic human malaria parasites, *P. falciparum* and *P. vivax*, have a distinct genome organization [30]. Indeed, the genome organization of these species appears to be shaped by clusters of parasite-specific gene families linked to pathogenicity, virulence, and parasite differentiation in the mosquito (Figure 2). The parasites employ the clustering of specific gene families to regulate the expression of stage-specific genes [44]. Identifying the molecular players controlling these critical assemblies and phenotypic differences will assist in uncovering new therapeutic targets.

### 2.2. Epigenetic Regulation of Gene Expression

Histone modifications’ spatial and temporal coordination impacts *Plasmodium* parasite differentiation by inducing, preventing, or poising transcriptional activation, DNA replication, damage repair, and chromatin condensation. Recent quantitative chromatin proteomic approaches and studies on the genome-wide localization of various epigenetic features have identified stage-specific histone modifications and their potential implication in the different life cycle stages of the malaria parasite [45,46] (Figure 2). Gene activation is associated with histone activation marks that include histone 3 Lysine 9 acetylation (H3K9ac), histone 3 Lysine 18 acetylation (H3K18ac), histone 3 Lysine 27 acetylation (H3K27ac), histone 3 lysine 4 trimethylation (H3K4me3), and the histone variant H2A.Z [43,47,48,49]. A study by Gupta et al. identified seven additional histone activation marks, including histone 4 Lysine 8 acetylation (H4K8ac), histone 4 Lysine 16 acetylation (H4K16ac), histone 4 tetra-acetylation (H4ac4), histone 3 Lysine 56 acetylation (H3K56ac), histone 3 Lysine 9 acetylation (H3K9ac), histone 3 Lysine 14 acetylation (H3K14ac), and histone 4 Lysine 20 methylation (H4K20me1) [50]. In contrast, gene repression is associated with distinct histone methylation and acetylation, including histone 3 lysine 9 trimethylation (H3K9me3) [51], histone 3 lysine 36 trimethylation (H3K36me3) [52,53], and histone 4 lysine 20 trimethylation (H4K20me3) [54].

Reversible gene silencing and activation are mediated by histone-modifying enzymes and histone-associated proteins that will “read, write or erase” histone code. *P. falciparum* is thought to encode 5 histone deacetylases (HDACs), 10 putative histone acetyltransferases (HATs), 10 methyltransferases (HMTs), and 3 demethylases [55,56]. These enzymes are predicted to add or remove methyl and acetyl groups to the lysine residues on histone molecules. Genetic studies have revealed several proteins implicated in gene silencing and heterochromatin state formation. The histone deacetylase enzyme *Pf*Hda2 [57], the sirtuin *Pf*Sir2a and *Pf*Sir2b [58], the H3K36-specific lysine methyltransferase *Pf*SET2 [52], and the H3K4-specific lysine methyltransferase *Pf*SET10 [59] are involved in the silencing of specific gene families, including sexual differentiation genes [54], and the virulence gene family known as var genes [52,54,57,58,59]. The var genes consist of approximately 60 paralogs of *P. falciparum* erythrocyte membrane protein 1 (*Pf*EMP1) located in the subtelomeric regions of multiple parasite chromosomes [60]. They are displayed on the surfaces of infected red blood cells and characterized by a monoallelic expression. The switch between antigenically and functionally distinct variants is responsible for antigenic variations and the cytoadherence of infected erythrocytes to the microvasculature [61].

Epigenomic and functional genetic studies have also identified several histone-associated proteins, the heterochromatin protein 1 (*Pf*HP1) [62,63] and some bromodomain-containing proteins (BDP), including *Pf*BDP1 [64], *Pf*BDP2 [64], and *Pf*BDP7 [65]. These proteins are implicated and required for silencing gene families involved in red blood cell invasion and gametocyte differentiation [62,64,65]. *Pf*HP1 is recruited by the H3K9me3 repressive epigenetic mark. The binding of *Pf*HP1 regulates the formation of heterochromatins. On the contrary, the histone post-translational modification histone H3, serine 10 phosphorylation (H3S10ph) impairs the binding of HP1 [66]. The authors suggested that H3 phosphorylation by Aurora B is part of a “methyl/phos switch” mechanism that displaces HP1 and perhaps other proteins from heterochromatin.

The activation of gene expression is mediated by the histone acetyltransferase *Pf*MYST (MOZ, YbF2, Sas2, Tip60-like) [67] and *Pf*GCN5 (General control non-depressible 5) [68]. An in vitro study assessing the activity of *P. falciparum*-purified MYST proteins revealed the recruitment of *Pf*MYST to acetylated histone 4 gene activation marks, suggesting *Pf*MYST implication in the expression of var genes and genes regulating asexual intraerythrocytic growth [67]. Similarly, functional studies on the divergent *Pf*GCN5 highlighted its association with euchromatin marks and the controlled expression of erythrocyte invasion and virulence genes [68].

Epigenomic studies have demonstrated that while gene activation marks (H3K9ac, H3K4me3) are widespread throughout the genome, the repressive marks (H3K9me3, H3K36me3, H4K20me3, HP1) appear to be specific for distinct gene families adjacent to subtelomeric var genes [51]. The gene subset comprises the antigenic variation gene families (var, rif, stevor), the Maurer’s cleft protein encoding two transmembrane domains (*Pf*mc-2tm), the nutrient acquisition multigene family (clag genes and acyl-CoA synthase genes), parasite differentiation genes (Apicomplexan Apetalla 2 genes; ApiAP2 genes), and the *Plasmodium* helical interspersed subtelomeric (PHIST) family.

Most epigenetic regulators and chromatin proteins implicated in gene silencing and activation in *Plasmodium* appear to be evolutionarily conserved in eukaryotic cells, suggesting a limitation in targeting them for therapeutic intervention. However, an in vitro screening of epigenetic modulator inhibitors against multiple life cycle stages of *P. falciparum* demonstrates the potential of using HDAC and HMT inhibitors as a new class of antimalarial drugs [69]. Coetzee et al. [69] tested the activity of compounds reported to be active against HDAC, HKM, HAT, and DNA demethylases in cancerous mammalian cells. Of the 95 compounds tested, 5 of them targeting histone acetylation and methylation showed potent multistage activity against asexual parasites, gametocyte stages, and the establishment of a successful infection in the mosquito. Thus, computer-aided drug design might be an alternative for developing a new class of antimalarial drugs targeting *Plasmodium* histone modification enzymes [70].

### 2.3. Transcriptional Regulation of Gene Expression

In addition to genomic organization and epigenetics, the regulatory machinery for gene expression in the *Plasmodium falciparum* includes the canonical TATA box binding proteins and the RNA-polymerase-II-dependent RNA expression [71]. A Hidden Markov Model (HMM) profile search for the *P. falciparum* genome identified 156 parasite-transcription-associated proteins, including all the 12 subunits of the RNA polymerase II and the general transcription factors (GTFs) for the basal transcription machinery [72]. Coulson et al. revealed that one group of GTFs, characterized by the histone 1 folding motif and their associated transcription factor complex (TFIID), is missing in the *P. falciparum* genome [72]. The authors suggested an evolutionary divergence of the basal transcriptional machinery in these parasites.

A low number of transcriptional regulation motifs and transcription factors that could account for the tight regulation of stage-specific gene expression patterns have been identified [71,72,73]. The ApiAp2 family, consisting of approximately 27 proteins, represents the main class of transcription factors known to regulate gene expression in *Plasmodium* [73,74]. These transcription factors contain the APETALA-2 (AP2) DNA binding domains and recognize multiple and distinct palindromic DNA sequences, including the TGCATGCA and GTGCAC motifs [75]. With some exceptions, most ApiAp2 transcription factors appear to regulate the expression of a unique gene set during one particular developmental stage [25,76,77,78,79,80]. Interestingly, the role of *Plasmodium* AP2 transcription factors is not restricted to regulating stage-specific gene expression. Indeed, the ApiAp2 proteins *Pf*AP2Tel [81] and *Pf*SIP2 [82], characterized by atypical AP2 domains that bind to specific DNA sequences, were shown to regulate telomere organization and promote gene silencing, respectively [81,82]. Additional transcription factors associated with gene expression that have been described include *Pf*Myb1 protein (a transcription factor belonging to the tryptophan cluster family) [83], *Pf*PREBP (a protein with four K-homology domains) [84], and *Pf*NF-YB (a protein containing a histone-fold domain) [85,86]. However, these transcription factors have only been associated with gene expression regulation in the intra-erythrocytic stage of the parasite [83,84,85,86].

Systematic profiling of the genome-wide occupancy of eighteen *Pf*ApiAp2 revealed that eight were preferentially associated with heterochromatic regions with differential coverage profiles [87]. Shang et al. [88] suggested that these heterochromatin-associated factors (*Pf*AP2-HFs) are likely part of the machinery recognizing repressive states in a DNA-motif-independent manner. Indeed, one of these *Pf*AP2-HFs was shown to be strictly recruited by *Pf*HP1 and to be a core component of heterochromatin.

The shortage of transcription and associated proteins strongly suggest an essential role for epigenetics, chromatin structure, genome organization, and posttranscriptional mechanisms in *Plasmodium* gene expression. Since the ApiAP2 protein family originated from the plant lineage and has no homologs in humans, they may be good antimalarial drug targets. An in silico screen to dock thousands of small molecules into the crystal structure of the AP2-EXP AP2 domain identified four compounds that specifically block DNA binding by AP2-EXP [89]. One of the inhibitors was shown to alter the transcriptome of *P. falciparum* trophozoite stages, characterized by a significant decrease in the abundance of AP2-EXP target genes. Two other DNA-binding inhibitory compounds have multi-stage anti-*Plasmodium* activity against blood and mosquito-stage parasites. A structure-based drug design combined with in vitro screening, in vivo studies in animal models, and the comprehensive profiling of ApiAp2 genetic polymorphisms in field-isolated parasites from natural human infection are likely to help in the development of a novel class of antimalarial drug targeting the *Plasmodium* ApiAp2 transcription factor. Furthermore, this compound class can potentially interfere with gene transcription in other Plasmodial species and apicomplexan parasites of medical importance, including *Toxoplasma gondii* and *Cryptosporidium* spp. While *T. gondii* causes many deaths in immunocompromised patients and fetal death when newly acquired during pregnancy [90], *Cryptosporidium* spp. Is one of the leading causes of infant diarrhea in developing countries [91]. *T. gondii* and *Cryptosporidium parvum* are predicted to encode 24 and 19 ApiAP2 genes, respectively, with some cross-conservation among Apicomplexan parasites [73,92].

## 3. Gametocyte Development

The development of male and female gametocytes within the vertebrate host and the sexual recombination in the mosquito vector are critical steps in the *Plasmodium falciparum* life cycle. These biological stages ensure parasite virulence and transmission. During the erythrocytic cycle, 1–5% of asexual parasites switch from the asexual multiplication pathway to a sexual differentiation pathway. The differentiation of sexually committed parasites into mature gametocytes is divided into five developmental stages (stage I–V) over 14 days. The differentiation process is characterized by dramatic alterations in parasite size, shape, and flexibility. Only early stages (stage I) and mature stage V are detectable in the peripheral blood, as the other stages are sequestered in the bone marrow and possibly other tissues [93]. The differentiation process underlying asexual to stage V gametocyte formation is governed by specific expression mechanisms extensively described in other reviews [94,95,96]. The following paragraphs summarize gene expression regulation mechanisms involved in asexual to early gametocyte formation (Table 1).

Several environmental and metabolic factors that trigger sexual commitment, including antimalarial drugs [97,98], lysophosphatidylcholine [99,100], host immune response [101,102], and parasite factors [103,104], have been reported. ApiAP2-G was the first transcription factor shown to govern the transcription of gametocyte-associated genes [24,25], and its regulation requires interaction with a second transcription factor, *Pf*AP2-I [105]. The cell surface receptors involved in signal transduction and the signaling cascade leading to the activation of ApiAp2-G transcription factors still need to be characterized.

While the *P. falciparum* genome is predominantly acetylated during the asexual developmental cycle, ApiAP2-G appears to be repressed, and its locus was shown to be associated with histone heterochromatin marks, including H3K4ac, H3K9me1, H3K36me2, *Pf*HP1, and the gametocyte essential factor *Pf*AP2-G5 [106]. *Pf*HP1-dependent gene silencing is antagonized by a sexual commitment activator protein called *Pf*GDV1 (*Plasmodium falciparum* gametocyte development 1) [107,108]. *Pf*GDV1 targets heterochromatin and triggers the eviction of *Pf*HP1. Interestingly, *Pf*GDV1 activation is, in turn, controlled by a multi-exon long non-coding *gdv1* antisense RNA (*Pfgdv1*asRNA) that initiates downstream of the *gdv1* locus [107].

The gametocyte essential factor AP2-G5 further prevents sexual commitment [108]. Indeed, *Pf*AP2-G5 binds upstream of the *Pf*ap2-g locus and other exogenic regions, hence suppressing ap2-g expression [108]. Although findings strongly suggest that ApiAP2 genes are under epigenetic control, histone post-translational modification linked to parasite conversion remains to be characterized. In addition, the signaling cascade events leading to the removal of the repressive state by *Pf*GDV1 and the activation of *Pf*ApiAP2-G remain unknown.

Sexual conversion can occur by two possible routes following specific histone modification marks that promote changes in gene expression and sexual differentiation [105,109]. One route suggests that the decision to undergo gametocyte development occurs during the asexual cycle preceding the erythrocyte reinvasion event that leads to gametocyte formation. In this model, the decision to undergo gametocyte development is linked to *Pf*AP2-G expression in the schizont of the preceding cycle [109]. The second route implies that the decision occurs within the same cycle, during the initial *Pf*AP2-G expression in ring stages [109]. Recently, Li et al. identified an ApiAP2 transcription factor in *P. yoelii*, AP2-O3, implicated in a gender-specific transcription program [110]. AP2-O3, expressed predominantly in female gametocytes, represses the expression of male-specific genes. Genetic studies have shown that the depletion of either HP1 or the histone deacetylase enzyme 2 (Hda2) resulted in the activation of *ap2-g* transcription and many heterochromatic genes [110].

Gametocyte development from stage I to stage V is characterized by euchromatic post-translational modifications and an abundance of repressive methylation marks on histone 3. Ealy gametocyte stages (I to III) are characterized by the H3K9me3, H3K27me2, H3K27me3, H3K36me2, H3K37me1, H3R17me1, and H3R17me2 modifications [46,106]. The role of arginine methylation as a key feature for the epigenetic regulation of gametocyte development and maturation was suggested by Von Grüning et al. [106].

Three-dimensional structure analyses of the *P. falciparum* genome revealed that the localization and interaction of sexual differentiation genes in a repressive center are critical for regulating sexual conversion [44]. Further studies are needed to unravel the machinery regulating genome localization and the signaling pathway governing sexual commitment. The repertoire of environmental triggers for sexual conversion is far from being exhaustive. Characterizing proteins interacting with the histone-associated protein *Pf*HP1 and *Pf*GDV1 will undoubtedly unravel the molecular player involved in parasite conversion.

**Table 1 microorganisms-11-01966-t001:** Summary of gene expression mechanisms involved in asexual to early gametocytes formation.

Mechanisms Underlying Sexual Conversion: from Asexual Forms to Gametocyte Stages	References	Studied Species
**1. Triggers**		
Environmental, metabolic, host, and parasite factors		
● Lysophosphatidylcholine	Brancucci N.M.B et al., 2017 [98]; Abdi A. et al., 2023 [99]	*P.f*
● Host immune response	Bruce M.C. et al., 1990 [100]; Nixon C.P. et al., 2018 [101];	*P.f*
● Drugs	Barkakaty B.N. et al., 1988 [97]; Buckling A. et al., 1999 [96]	*P.f*
● Parasite factors	Ayanful-Torgby R. et al., 2016 [102]; Chawla J. et al., 2023 [103]	*P.f*
**2. Genome organization**		
● Sexual differentiation genes are clustered in a repressive center	Bunnik J.L. et al., 2018 [29]	*P.f*
**3. Transcriptional regulation**		
● Transcription factor regulating sexual conversion: ApiAP2-G, ApiAp2-I	Sinha A. et al., 2014 [24]; Josling G.A. et al., 2020 [104]	*P.f and P.b*
● Transcription factor regulating gender: AP2-O3 (male-gene repressor expressed in female gametocytes)	Li Z. et al., 2021 [109]	*P.y*
**4. Epigenetic regulation**		
❖ **Before commitment**		
● Histone silencing marks: H3K4ac, H3K9me1, H3K36me2	Jiang L. et al., 2013 [51]; von Gruning H. et al., 2022 [105]	*P.f*
● Histone modification enzymes: Hda2	Coleman B.I. et al., 2014 [56]	*P.f*
● Histone-associated proteins for gene silencing: HP1, AP2-G5	Flueck C. et al., 2009 [61]; Shang X. et al., 2021 [107]; von Gruning H. et al., 2022 [105];	*P.f*
● Regulatory antisense RNA: *gdv*1asRNA	Filarsky M. et al., 2018 [106]	*P.f*
❖ **Early-stage gametocytes (stage 1 to stage 4)**		
● Histone marks for the dissociation of repressive protein HP1: H3S10ph	Hirota T. et al., 2005 [65]	*-*
● Histone activation marks: H3K9me3, H3K27me2, H3K27me3, H3K36me2, H3K37me1, H3R17me1, and H3R17me2	von Gruning H. et al., 2022 [105];	*P.f*
● Histone modification enzymes: presumably MYST and GCN5	Miao J. et al., 2010 [66]; Miao J. et al., 2021 [67]	*P.f*
● Histone-associated proteins for: GDV1 (sexual commitment activator)	Filarsky M. et al., 2018 [106]	*P.f*
**GAPS:**		
● Machinery regulating genome localization remains to be characterized		
● Repertoire of environmental triggers far from being exhaustive		
● Signaling pathway governing sexual commitment remains to be thoroughly characterized		

Several environmental and metabolic factors trigger sexual commitment. The differentiation of sexually committed parasites into gametocytes is characterized morphologically by dramatic alterations in parasite size, shape, and flexibility. Specific expression mechanisms govern the differentiation. Histone deacetylase 2 (Hda2); Heterochromatin protein 1 (HP1); General control non-depressible 5 (GCN5); MYST (MOZ, YbF2, Sas2, Tip60-like), Apicomplexan Apetalla 2 (ApiAP2); Gametocyte development 1 (GDV1); antisense RNA (asRNA).

Despite their similitude with human enzymes, histone-modifying enzymes could be potential targets for developing antimalarial drugs with transmission-blocking activities. Stenzel et al. [111] designed, synthesized, and tested the biological activity of thirteen terephthalic-acid-based HDAC inhibitors. A subset of these compounds had moderate activity against *P. falciparum* gametocytes but showed sub-micromolar transmission-blocking activity against the rodent malaria parasite *P. berghei*. Hence, histone-modifying enzymes are potential targets to prevent malaria infection in the mosquito.

## 4. Gamete Development

Establishing a successful malaria infection in the *Anopheles* mosquito and the subsequent spread of competent parasites depend on the coordinated development of the *Plasmodium* parasite in the mosquito midgut. The differentiation process in the mosquito midgut, lasting almost 20 h, is defined by the rapid conversion of mature stage V female and male gametocytes into female macrogametes and male microgametes, respectively [6,10,11]. This step is followed by the sexual reproduction between macro- and microgametes and, finally, the conversion of newly formed zygotes into ookinetes [6,11]. The transition in the mosquito midgut constitutes a significant bottleneck in the *Plasmodium* life cycle as there is a 316-fold loss in parasite abundance from the gametocyte to ookinete stage and a 100-fold loss from the ookinete to oocyst stage [112]. Due to this considerable reduction in the parasite population, the plasmodial midgut stages are attractive targets for developing transmission-blocking interventions. Hence, malaria parasite developmental stages in the mosquito midgut have been extensively studied, and several key mechanisms and proteins orchestrating the differentiation process have been described, providing potential targets for vaccines and drug development [113].

Gamete formation is morphologically characterized by a rounding-up of the cell, followed by the parasitophorous vacuole membrane (PVM) rupture and parasite egress from the erythrocyte [10,114]. The differentiation of male gametocytes into male gametes involves three successive mitotic DNA replications that will produce eight motile microgametes through a process called exflagellation. Upon completion of DNA replication, the microgametes egress from the host cell [10]. In contrast, the differentiation of female gametocytes results in a rounding-up of the parasite with no DNA replication and the emergence of the forming gamete from the infected red blood cell [6]. Specific expression mechanisms govern the differentiation of mature stage V gametocytes into gametes (Table 2).

### 4.1. Gene Expression Regulation Controlling Gametogenesis

Transcriptomic analysis of *P. falciparum* gametocytes and gametes revealed that small transcriptome changes characterize gametogenesis compared to other life-stage differentiation processes [115]. Most studies investigating gamete to sporozoite formation rely on *Plasmodium berghei,* causing rodent malaria. The rodent parasites are not of direct practical concern to humans. However, the easy access to *Plasmodium* life stages in the mosquito and the practicability for the experimental study of human malaria made them a good model for human malaria.

In *Plasmodium berghei*, the change in gene expression necessary for female gametogenesis is determined by the translation of a large number of mRNAs maintained in a repressive translational state [116,117]. Indeed, in female gametocytes, many transcripts are synthesized but translationally repressed until needed for macrogamete formation and zygote-to-ookinete transformation. Using the *Plasmodium berghei* model system, Guerreiro et al. revealed that approximately 50% of the transcriptome is maintained in this translationally repressed state by a messenger ribonucleoprotein (mRNP) composed of 16 major factors, including the RNA helicase DOZI (development of zygote inhibited) and the Sm-like factor CITH (homolog of worm CAR-I and fly Trailer Hitch) [118]. Using an in vitro translation assay, Tarique et al. characterized *P. falciparum* DOZI (*Pf*DZ50) and demonstrated that the protein inhibits translation [119], suggesting a function similar to that reported for *P. berghei* female gametogenesis.

The expression of stage-specific genes that are necessary for microgamete formation is regulated by transient and reversible protein phosphorylation followed by the de novo synthesis of genes involved in DNA replication, axoneme assembly and motility, chromatin condensation, cytokinesis, and exflagellation [120,121]. Proteomic studies in *P. berghei* gametes revealed that proteins implicated in RNA translation, protein biosynthesis, glycolysis, environmental stress response, and tubulin-associated cytoskeleton dynamics are predominantly regulated during gamete formation [122]. Gene expression and protein synthesis regulation during both male and female gametogenesis depend highly on the signaling cascade orchestrated by stage-specific kinases and phosphatases [120,123,124].

### 4.2. Signaling Cascade Controlling Gametogenesis

Gametocyte activation is stimulated by environmental stimuli, including a temperature drop by approximately 5 °C, the presence of mosquito-derived xanthurenic acid (XA), and an increase in extracellular pH from 7.2 to about 8 [125,126,127]. *Plasmodium* receptors responsible for sensing temperature drops and binding to XA have not been identified yet. The exposure of *P. falciparum* gametocytes to XA has been shown to increase cyclic GMP (cGMP) levels in the gametocyte, suggesting parasite guanylyl cyclase (GC) activation [128]. In the rodent malaria parasites, the increase in cGMP eventually leads to an increase in the cytoplasmic calcium level necessary to trigger a calcium-dependent stage-specific gene expression and regulation pathway [129,130,131].

The *Plasmodium falciparum* genome encodes two guanylyl cyclase proteins (GCα and GCβ) that are only expressed in sexual-stage parasites. Functional studies in *P. falciparum* revealed that these proteins are active guanylyl cyclases and likely important in gametocyte activation [132]. Pharmacological and genetic studies investigating the signaling cascade of *P. falciparum* and *P. berghei* gamete formation have shown that GC activation leads to the formation of a secondary messenger cyclic GMP (cGMP), which in turn activates the protein kinase G (PKG), the unique effector identified in *Plasmodium* [128,133]. Taylor et al. revealed that the tight regulation of cGMP concentration is critical for *P. falciparum* gametocyte conversion and that premature high cGMP levels are deleterious for gamete formation [134]. Bennink et al. suggested that activation of PKG leads to the hydrolysis of phosphatidylinositol-(4,5)-bisphosphate (PIP2) by the phosphoinositide-specific phospholipase C (PI-PLC) and the production of two secondary messengers, diacylglycerol (DAG) and inositol triphosphate (IP3) [135,136]. Although IP3 induces the opening of calcium channels on the *Plasmodium* endoplasmic reticulum membrane, no ortholog of an IP3 receptor channel has yet been identified. A recent biochemical approach of an IP3 affinity chromatography column combined with bioinformatics has revealed a potential transporter associated with multidrug resistance in *P. falciparum* [137]. Additional work will elucidate the function of this novel protein.

The rapid release of calcium in mature gametocytes mediates a stage-specific calcium-dependent effector pathway in all gametogenesis steps, including gamete formation, gamete egress, microgamete mitotic maturation, and exflagellation [128,133,135]. In *Plasmodium*, the intracellular calcium level is sensed by calcium-dependent protein kinases (CDPKs) [138]. Macro- and microgamete-specific CDPKs that regulate the expression and activation of genes necessary for converting gametocytes into gametes have been identified [121,139,140].

In mature female gametocytes, the increase in cytoplasmic Ca^2+^ is sensed by CDPK1 [139]. CDPK1 has been identified as the key protein that regulates the translation of mRNAs in a temporal and stage-specific manner during macrogamete formation [139].

The differentiation of male gametocytes is regulated by a male-specific calcium-dependent protein kinase CDPK4, which initiates DNA replication, axoneme assembly, and cell motility [121,141]. In *P. berghei,* the lysis of the host cell membrane(s) surrounding the microgametocyte is mediated by CDPK1 [139]. In addition, CDPK1 translationally activates mRNA species in the developing zygote that remain repressed in macrogametes [139]. The cell-division cycle protein 20 (CDC20) appears to regulate male gametocyte mitotic division but not the one during schizogony [142]. Whether CDC20 works with CDPK1 and anaphase-promoting complex 3 (APC3) to modulate chromosome condensation and cytokinesis for microgamete formation is not yet determined [143]. In *P. berghei*, a histone chaperone protein termed FACT-L (facilitates chromatin transcription), which facilitates chromatin transcription, was shown to be involved in male gametocyte DNA replication and the production of fertile microgametes [144]. The formation of the flagella requires the formation of basal bodies and the assembly of axonemes.

Upon completion of DNA replication, axonemes become motile, facilitating the egress of the microgametes from the host cell. Genetic studies in *P. berghei* have revealed that axoneme assembly and motility require several proteins, including the stage-specific Actin II [145], the armadillo-repeat motif protein *Pf*16 [146], the spindle-assembly-related protein SAS6 [147], the SR protein kinase (SRPK) [148], and a gametocyte-specific mitogen-activated protein kinase 2 (MAP2) [149,150]. The *Plasmodium falciparum* kinome includes four NIMA-related kinases (*Pf*NEK 1 to 4). While *Pf*NEK-1 is expressed in asexual and sexual stages, the mRNA transcripts of *Pf*NEK-2, *Pf*NEk-3, and *Pf*NEK-4 are exclusively expressed in gametocytes. NEK1 and NEK3 have been shown to activate the atypical MAP2 protein through phosphorylation [151,152]. *Pf*NEK-2 and *Pf*NEK-4 are required for meiosis completion in the ookinete [153,154]. A metallo-dependent protein phosphatase, PPM1, also plays an important role in *P. berghei* male gametocyte exflagellation [124].

Kinases and phosphatases involved in gamete formation can be considered promising targets for drug development. A high-throughput screening of *P. falciparum*-cGMP-dependent protein kinase identified a thiazole scaffold that kills erythrocytic and sexual-stage parasites [155]. Since the malarian PKG differs from the mammalian PKGs, this scaffold represents a good starting point for developing a new class of antimalarial drugs. Recently, Xitong et al. [156] assessed the activity of 25 phosphatase inhibitors against *Plasmodium berghei* sexual development and transmissibility to the mosquito. Two compounds from the panel effectively inhibited different development stages, from gametogenesis to ookinete maturation. These examples highlight that in silico modeling and screening combined with in vivo and ex vivo approaches using mouse models and human malaria could help identify parasite kinase and phosphatase inhibitors. The kinases regulating male gamete exflagellation (CDPK4 and atypical MAP-2) and DNA replication (Nek-2, Nek-4) should be of great interest in the screening.

Although rodent malaria parasites are sound model systems to understand *P. falciprum* gametogenesis, more studies relying on human malaria parasites are necessary to grasp the complexity of the early stages of malaria transmission. Those studies should consider including field studies to better understand the biology of transmission during natural infection.

**Table 2 microorganisms-11-01966-t002:** Summary of gene expression and signaling cascades controlling gametogenesis.

Mechanisms Underlying the Differentiation from Mature Stage V Gametocytes to Gametes	References	Studied Species
**1. Gene expression characteristics**		
● Small transcriptome changes between the gametocyte and the gametes	Ngwa C.J. et al., 2013 [114]	*P.f*
● Gene expression and protein synthesis regulation dependent on signaling cascade orchestrated by kinases and phosphatases	Invergo B.M. et al., 2017 [119]	*P.b*
**2. Triggers**		
● Temperature drop by 5 °C and extracellular pH of ~8	Billker O. et al., 1997 [124];	*P.b*
● Xanthurenic acid	Garcia G.E. et al., 1998 [125]	*P.f and P.g*
**3. Signaling pathway**		
● Stimuli lead to increase in cGMP and the activation of PKG;	Mc Robert L. et al., 2008 [127]; Brochet M. et al., 2021 [128]	*P.f, P.b, P.y*
● PKG leads to the release of Ca^2+^ from the parasite endoplasmic reticulum and the activation of CDPKs	Wang P.P. et al., 2022 [132]	*P.b*
**4. Female gametogenesis**		
● mRNAs maintained in a repressive translational state by messenger ribonucleoproteins (DOZI+~16 factors + Sm-like factor CITH)	Mair G.R. et al., 2010 [116]; Tarique M. et al., 2013 [118]; Guerreiro A. et al., 2014 [117];	*P.b and P.f*
● mRNA translation regulator: CDPK1	Sebastian S. et al., 2012 [138]	*P.b*
● Egress from RBC: CDPK1	Invergo B.M. et al., 2017 [119]	*P.b*
**5. Male gametogenesis**		
● Transient and reversible protein phosphorylation and de novo synthesis of effectors involved in DNA replication, axoneme assembly and motility, chromatin condensation, cytokinesis, and exflagellation	Billker O. et al., 2004 [120]; Invergo B.M. et al., 2017 [119]	*P.b*
● DNA replication, axonemes assembly, and cell motility regulation: CDPK4	Invergo B.M. et al., 2017 [119]; Kumar S. et al., 2021 [140]	*P.b and P.f*
● Mitotic division regulation: CDPK1, CDC20, APC3, and FACT-L	Guttery D.C. et al., 2012 [141]; Invergo B.M. et al., 2017 [119]; Wall R.J. et al., 2018 [142]	*P.b*
● Axoneme assembly regulation: Actin II, *Pf*16, SAS6, SRPK, MAP2, NEK1, NEK3	Dorin D. et al., 2001 [150]; Lye Y.M. et al., 2006 [151]; Straschil U. et al., 2010 [145]; Deligianni E. et al., 2011 [144]; Marques S.R. et al., 2015 [146]; Invergo B.M. et al., 2017 [119]	*P.f and P.b*
● Exflagellation: protein phosphatase, PPM1	Guttery D.C. et al., 2014 [123]	*P.b*
● Egress from RBC: CDPK1	Invergo B.M. et al., 2017 [119]	*P.b*
**GAPS:**		
● Plasmodium-sensing receptors at the cell surface are unknown		
● IP3 receptor channel at the ER membrane has yet to be identified		
● Relatively few studies used *P. falciparum*		

Gene expression and protein synthesis regulation during both male and female gametogenesis depend highly on signaling cascades orchestrated by stage-specific kinases and phosphatases. Protein kinase G (PKG); Calcium-dependent protein kinases (CDPKs); Development of zygote inhibited (DOZI); Cell Division Cycle 20 (CDC20); FACT-L (facilitates chromatin transcription L); Armadillo-repeat motif protein (Pf16); Spindle-assembly-related protein 6 (SAS6); SR protein kinase (SRPK); Mitogen-activated protein kinase2 (MAP2); NIMA-related kinases (NEKs); Metallo-dependent protein phosphatase (PPM1).

## 5. Zygote to Ookinete Development

### 5.1. Gamete Fusion and Zygote Formation

The zygote results from the fusion between a fertile male microgamete and a female macrogamete. Fertilization of a macrogamete by a microgamete is mediated by stage- and sex-specific proteins synthesized in the respective gamete before the fusion event (Table 3). The proteins involved in gamete fusion also mediate the prerequisite recognition and attachment steps. Genetic studies of *P. falciparum* and *P. berghei* genes have revealed that P48/45 and P230, two members of a protein family defined by a disulfide bonding pattern of six conserved cysteine residues, are essential for male gamete fertility and fusion with macrogametes [157,158,159]. In *P. falciparum,* P48/45 and P230 are localized on the gamete surface and have been shown to form a complex necessary for the fusion of the microgamete to the macrogamete. In vitro and in vivo studies in the mouse model have demonstrated the critical implication of P48/45 and P230 in fertilization. These findings led to their use as targets for transmission-blocking vaccines [113,160]. In macrogamete *Pf*s47, a paralog of P48/45 was identified [161]. Functional studies demonstrated the expression of *Pf*s47 exclusively on the surface of female gametes, but the protein did not appear crucial for fertilization [161].

**Table 3 microorganisms-11-01966-t003:** Summary of the regulation mechanism in gamete fusion.

Mechanisms Underlying Fertilization	References	Studied Species
**1. Characteristics**		
Gamete fusion is mediated by stage- and sex-specific proteins synthesized in the respective gametes before the fusion event		
**2. Proteins mediating fusion**		
● Gamete surface proteins: P48/45 (male gamete fertilization factor) and P230	Rener J. et al., 1983 [157]; van Dijk, M.R. et al., 2001 [158]; Williamson K.C. et al., 2003 [156]	*P.f. and P.b.*
● Other factors: HAP2/GCS1, histone chaperone protein FACT	Lui Y. et al., 2008 [161]	*P.b.*
**GAPS:**		
● Are all the players involved in the fusion event determined?		
● Is the quaternary structure of the fusion complex fully determined?		

Fertilization of a macrogamete by a microgamete is mediated by stage- and sex-specific proteins synthesized in the respective gamete before the fusion event. Hapless 2/Generative cell-specific 1 (HAP2/GCS1); facilitates chromatin transcription (FACT).

The evolutionarily conserved class II gamete fusogen HAP2/GCS1 (hapless 2/Generative-cell-specific 1) has also been implicated in the fusion of the plasma membranes of two haploid gametes [162]. Gene disruption studies of *p48/45* or *hap2/gcs1* resulted in sterility due to the inability of the male gamete to either attach or fuse to fertile female gametes [159,162]. Studies investigating gamete fusion in *Plasmodium* revealed that a histone chaperone protein named FACT (facilitates chromatin transcription) plays an essential role in fertilization. However, the mechanism of action of this nuclear protein is still unclear [144].

The fusion of the two gametes requires the prior disassembly of a peculiar organelle located underneath the parasite plasma membrane and is named the inner membrane complex (IMC) [5]. The IMC will be reassembled later to ensure the pellicle’s integrity and parasite polarity [5,163]. The IMC is a membranous scaffold in the Alveolata, a group of diverse unicellular eukaryotes, including *Plasmodium* spp., *Toxoplasma gondii*, the ciliates, and dinoflagellates. This membranous patchwork is anchored to the subpellicular microtubule network. It also interacts with various parasite-specific proteins to coordinate parasite morphological changes, the segmentation of daughter cells during asexual replication, and parasite motility. Interestingly, the localization and interaction of many IMC proteins are regulated by post-translational S-palmitoylation mediated by the palmitoyl-acyl-transferase DHHC2 (the ortholog of *Pf*DHHC1) [164]. Inhibitor studies revealed that DHHC2 palmitoylation is critical to zygote differentiation during the initial mosquito infection with *P. berghei* [165]. Hence, stage-specific palmitoylation enzymes could be novel targets for disrupting IMC assembly and zygote formation and differentiation into ookinetes.

The concept of transmission-blocking immunity is mostly antibody-mediated. Therefore, the development of transmission-blocking vaccines (TBVs) focuses on inducing potent antibodies sustained at adequate levels. Extensive efforts toward the clinical development of *P. falciparum* TBVs are undertaken, but the functional activities associated with most antibodies remain modest. Further study must be conducted to determine the players involved in gamete fusion and their association with each other to orchestrate gamete fusion. The disassembly of the IMC is a pre-requisite for gamete fusion, and targeting the enzymes modulating this biological process could be an approach to interfere with the establishment of the infection in the mosquito gut. *Plasmodium falciparum* is predicted to encode 12 putative palmitoyl acyl-transferases thought to ensure lipid-based palmitoylation of parasite proteins and act as a biological rheostat for protein–protein interactions and subcellular trafficking [166]. Inhibitors of palmitoylation enzymes could constitute a new class of antimalarial drugs targeting multiple parasite life stages and other apicomplexan parasites of medical importance [166,167,168].

### 5.2. Molecular and Genetic Mechanisms of Fertilization

Gene expression studies in *P. berghei* revealed that zygote development and differentiation into ookinetes require the transcriptional activation of several maternal silenced mRNAs in the zygote. A study investigating the parental contribution of transcripts implicated in zygote differentiation revealed that while inherited maternal mRNAs are activated to drive the early stage of zygote differentiation, the paternal alleles are initially silenced and then reactivated [19,169].

The derepression of a maternal transcript post fertilization will drive significant morphological changes defined by the formation of the IMC and the secretory organelles, as well as the elongation and apical polarization of the differentiating zygote [5,170,171]. The dynamic organization of the IMC and the parasite cytoskeleton is essentially coordinated post-translationally by reversible phosphorylation [139,148,172] and palmitoylation [165,173]. A systematic analysis of the *Plasmodium* kinome, combined with genetic studies in *P. berghei,* revealed that protein kinase 7 (PK7) and cyclin-G-associated protein (GAK) are essential in ookinete formation [148,172]. GAK is predicted to regulate clathrin-mediated vesicle trafficking and membrane fusion, suggesting its involvement in the formation of the secretory organelles or the assembly of the IMC [148]. PK7 mutants were blocked in ookinete formation, but the mechanism by which this kinase regulates ookinete development remains undetermined. Reverse genetics studies revealed that the *Plasmodium* phosphatase PPKL (protein phosphatase with kelch-like domains) is essential during ookinete differentiation and is involved in defining ookinete polarity, pellicle morphology and integrity, and ookinete motility [174]. *Plasmodium falciparum* DHHC1, a palmitoyl-S-acyl-transferase (PAT) containing a conserved DH(H/Y)C motif, was shown to be exclusively localized to the IMC [165,173]. DHHC1 is apicomplexan-specific and was implicated in ookinete formation and morphogenesis [165,173].

Subsequent regulation of stage-specific gene expression requires a de novo synthesis of ookinete-specific genes by the transcription factor ApiAp2-O [175]. ApiAp2-O appears to be associated with more than 500 genes involved in ookinete development, motility, midgut invasion, and parasite escape from mosquito immunity [78]. The ookinete-specific genes mediate nuclear fusion in the diploid zygote, DNA replication, and meiosis that will produce a motile tetraploid ookinete [176]. The tetraploid state was shown to persist throughout the ookinete stage until the formation of sporozoites in the oocyst [176,177]. Like in eukaryotic model systems, meiosis and cell cycle progression in the *Plasmodium* parasite are regulated by NIMA-related kinases (Neks). Gene disruption studies in *P. berghei* revealed that Nek-4 and Nek-2 are abundantly expressed during the gametocyte stages. Nek-4 and Nek-2 are essential for zygote differentiation into the ookinete stage but not for gamete formation and fertilization [153,154]. Given the success in developing drugs targeting human kinases, *Plasmodium* kinases are attractive targets for the next generation of antimalarials. Indeed, ongoing efforts attempt to characterize *Plasmodium* kinases while evaluating them as antimalarial drug targets [178,179].

### 5.3. Formation and Maturation of Ookinete

Ookinete maturation is completed 19 to 36 h post-gametocyte ingestion in the blood meal. The ookinete will quickly exit the midgut lumen via an intracellular or intercellular route [180]. Ookinete motility is regulated by kinases. The cGMP-dependent protein kinase (PKG) pathway activates the gliding motility apparatus on the IMC, and the calcium-dependent protein kinase 3 (CDPK3) mobilizes the intracellularly stored calcium necessary for signaling [181,182]. The traversal of the midgut wall is mediated by an impressive set of microneme proteins discharged by the mature ookinete [183]. Proteomic [183] and genetics studies identified key proteins for midgut traversal, including the circumsporozoite-TRAP-related protein CTRP [184], the membrane-attack ookinete protein (MAOP) [185], the secreted ookinete adhesive protein SOAP [186], the von-Willebrand-Factor-A-domain-related protein WARP [187], the cell-traversal protein for ookinetes and sporozoites CelTOS [188], and the chitinase 1 (CHT1) [189,190]. A summary of the molecular and genetic mechanisms regulating zygote differentiation into ookinetes is provided in Table 4.

The ookinete stage is an attractive target for transmission-blocking strategies as the mosquito immune system naturally kills many ookinetes. Ookinete proteins involved in the attachment and invasion of the midgut epithelial cells are potential vaccine targets. Two ookinetes surface proteins, P25 and P28, that play a role in ookinete adhesion to the midgut and differentiation into oocyst are candidates for transmission-blocking vaccines [113]. Transmission-blocking strategies comprise gametocytocidal drugs, transmission-blocking vaccines (TBV), and the engineering of genetically modified mosquitos refractory to *Plasmodium* infection [191]. The rationale for transmission-blocking drugs is to promote gametocyte clearance in the human host and to block onward parasite transmission by targeting blood stage parasites [192]. On the contrary, transmission-blocking vaccines aim to induce, in the human host, the production of antibodies against specific proteins accessible in the mosquito gut to prevent mosquitoes from carrying and spreading the parasites. Targeted proteins include surface proteins of *Plasmodium* stages found in the mosquito [113] or mosquito-specific effectors implicated in the infection [193]. Antibodies will be ingested by the mosquitoes during their blood meal and are expected to interact with the target in the mosquito gut before the ookinete transversal of the midgut wall. Therefore, essential ookinete surface proteins and potentially micronemal proteins secreted by the parasite at the time of gut lumen exit could be targeted by TBV approaches.

**Table 4 microorganisms-11-01966-t004:** Summary of the molecular and genetic mechanisms regulating zygote differentiation into ookinete.

Mechanisms Underlying Zygote to Ookinete Formation	References	Studied Species
**1. Characteristics**		
Zygote development and differentiation into ookinete require the transcriptional activation of several maternal silenced mRNAs in the zygote	Akinosoglou K.A. et al., 2015 [18]; Ukegbu C.V. et al., 2015 [168]	*P.b.*
**2. Gene expression mechanisms**		
● Derepression of inherited maternal mRNAs drive the early stage of zygote differentiation	Akinosoglou K.A. et al., 2015 [18]	*P.b.*
● De novo synthesis of ookinete-specific genes	Janse C.J. et al., 1986 [175]	*P.b.*
**3. Transcriptional activation of maternal mRNAs control zygote morphological changes**		
● Assembly of the subpellicular membrane complex (IMC) and the subpellicular cytoskeleton	Kono M. et al., 2012 [4]; Volkmann K. et al., 2012 [170]; Poulin B. et al., 2013 [170]	*P.b.*
● Formation of secretory organelles	Frenal K. et al., 2013 [165]; Kaneko I. et al., 2015 [77]	*P.b.*
● Cell polarization and morphology changes	Kono M. et al., 2012 [4]; Volkmann K. et al., 2012 [169]; Poulin B. et al., 2013 [170]	*P.b.*
● Regulatory proteins coordinating morphological changes: Processes essentially coordinated post-translationally by reversible phosphorylation and palmitoylation:		
● Protein kinase 7	Dorin-Semblat D. et al., 2008 [171]; Tewari R. et al., 2010 [147]	*P.f. and P.b.*
● Cyclin-G-associated protein (GAK)	Tewari R. et al., 2010 [147]	*P.b.*
● Phosphatase PPKL	Guttery D.S. et al., 2012 [173]	*P.b.*
● Palmitoylation enzymes DHHC1	Wetzel J. et al., 2015 [172]; Santos J.M. et al., 2015 [164]	*P.f. and P.b.*
**4. Regulation of zygote-specific gene expression**		
● Stage-specific transcription factor: ApiAP2-O	Yuda M. et al., 2009 [174]; Kaneko I. et al., 2015 [77]	*P.b.*
**5. Meiosis and cell cycle progression regulators**		
● NIMA-related Kinases (NEK4 and NEK2)	Reininger L. et al., 2005 [153]; Reininger L. et al., 2009 [152]	*P.f. and P.b.*
**6. Ookinete maturation**		
● Motility regulation: PKG, CDPK3	Ishino T. et al., 2006 [181]; Moon R.W. et al., 2009 [180];	*P.b.*
● Midgut wall traversal via micronemal proteins discharge: CTRP, MAOP, SOAP, WARP, CelTOS, CHT1	Dessens J.T et al., 1999 [183]; Vinetz J.M. et al., 2000 [188]; Yuda M. et al., 2001 [186]; Dessens J.T. et al., 2003 [185]; Kadota K. et al., 2004 [184]; Kariu t. et al., 2006 [187]; Viswanath V.K. et al. 2021 [189]	*P.b., P.g*
**7. Ookinete differentiation**		
● Triggers: extracellular matrix	Kaslow D.C. et al., 1994 [193]	*-*
● Midgut wall traversal: CTRP, MAOP, SOAP, WARP, CelTOS, CHT1		*P.b., P.g*
**GAPS:**		
● Mechanisms triggering ookinete to oocyst development are still unclear		
● Relatively few transcriptomics and proteomics studies assessing changes associated with the early stage of oocyst development		
● Gene regulation mechanisms are understudied		
● Role of abundant antisense transcripts in ookinetes remains to be elucidated		

Zygote development and differentiation into ookinetes first require the transcriptional activation of several maternal silenced mRNAs in the zygote. Subsequent regulation of stage-specific gene expression requires de novo synthesis of ookinete-specific genes. Inner Membrane Complex (IMC); Protein kinase 7 (PK7); Cyclin G-associated protein (GAK); Protein phosphatase with kelch-like domains (PPKL); palmitoyl-S-acyl-transferase with conserved DH(H/Y)C motif (DHHC1); NIMA-related kinases (NEK); Apicomplexan Apetalla 2 (ApiAP2); Protein kinase G (PKG); Calcium-dependent protein kinase 3 (CDPK3); Circumsporozoite-TRAP-related protein CTRP; Membrane-attack ookinete protein (MAOP); secreted ookinete adhesive protein (SOAP); von-Willebrand-Factor-A-domain-related protein (WARP); cell-traversal protein for ookinetes and sporozoites (CelTOS); Chitinase 1 (CHT1).

As mentioned above, parasite-specific kinases, whose functions are required for parasite motility, are also good targets for *Plasmodium* multi-stage drugs.

## 6. Ookinete to Oocyst Development

Following the traversal of the midgut epithelium, ookinetes settle in the basal lamina surrounding the gut and differentiate into oocysts. Ookinetes can be found in the basal lamina 18–24 h after an infective blood meal [12]. The mechanisms triggering the differentiation process are still unclear, but the extracellular matrices composed of collagen, fibronectin, laminin, and chondroitin sulfate are thought to play a role [194]. With the current proteomic and genomic tools, future studies should further investigate mosquito factors involved in ookinete differentiation.

Compared with other developmental stages, relatively few transcriptomics and proteomics studies assess changes associated with the early stage of oocyst development. Sequencing strand-specific cDNA libraries of seven *Plasmodium* stages has revealed an abundance of antisense transcripts in gametocytes and ookinetes [195]. These findings suggest that antisense RNA plays a role in gene expression regulation in sexual stages. However, the roles of these antisense transcripts still have to be investigated [195]. Studies investigating the ookinete to oocyst differentiation are required to understand these processes better.

## 7. Conclusions

Functional studies have uncovered critical processes associated with *Plasmodium* sexual differentiation within the vertebrate host and the mosquito vector. During intraerythrocytic development, parasites appear to predominantly regulate the expression of functional gene products in a stage-specific manner via epigenetic and transcriptional regulation. When preparing to change hosts, gametocyte formation necessitates post-translational repression mechanisms and epigenetic and transcriptional regulation. The fast differentiation process within the mosquito gut relies on the post-translational modification of proteins by specific kinases, phosphatases, and lipid modifications.

The list of the molecular players involved in *Plasmodium* sexual differentiation is far from complete. The development of new approaches, including single-cell technologies, will provide greater resolution on transcriptional, epigenomic, and metabolomic variations that regulate parasite differentiation. Current findings suggest that functional genetics data need to be integrated into studies assessing the function of malaria proteins for a better understanding of the developmental processes. Although rodent malaria parasites are good model systems to understand *P. falciparum* development within the mosquito, more studies relying on human malaria parasites are necessary to grasp the complexity of malaria transmission and infection in the mosquito. *Plasmodium* functional studies have been carried out using parasites maintained and propagated in vitro. With environmental factors affecting considerable parasite gene expression, additional epigenomic, proteomics, transcriptomic, and metabolomic studies from field isolates are pertinent to better understand the processes that govern parasite differentiation. Finally, these studies will have to target all six malaria species infecting humans to identify conserved and species-specific processes that could be used to discover new vaccine candidates and novel classes of antimalarial drugs. Computer-aided approaches and in silico modeling might be alternatives for developing new classes of antimalarial drugs targeting parasite enzymes.

## Figures and Tables

**Figure 1 microorganisms-11-01966-f001:**
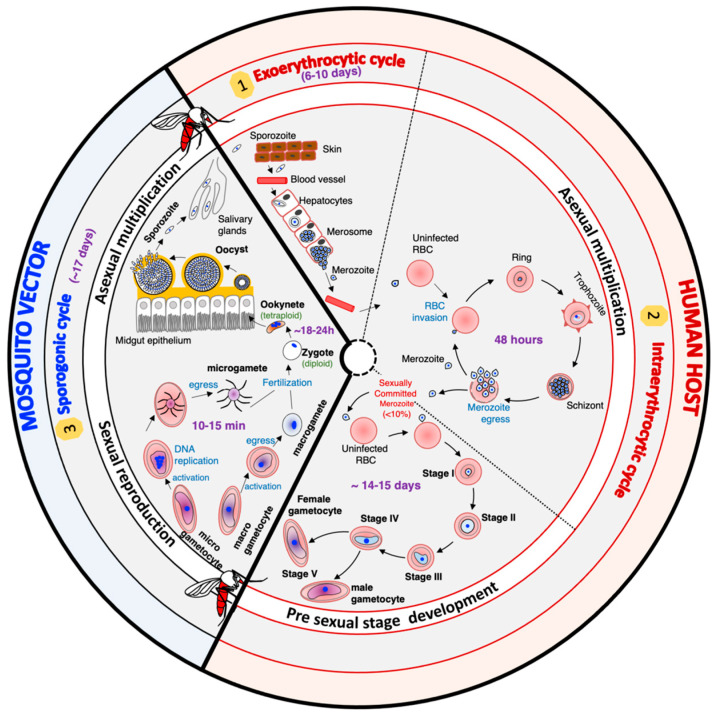
The life cycle of *Plasmodium falciparum* parasites.

**Figure 2 microorganisms-11-01966-f002:**
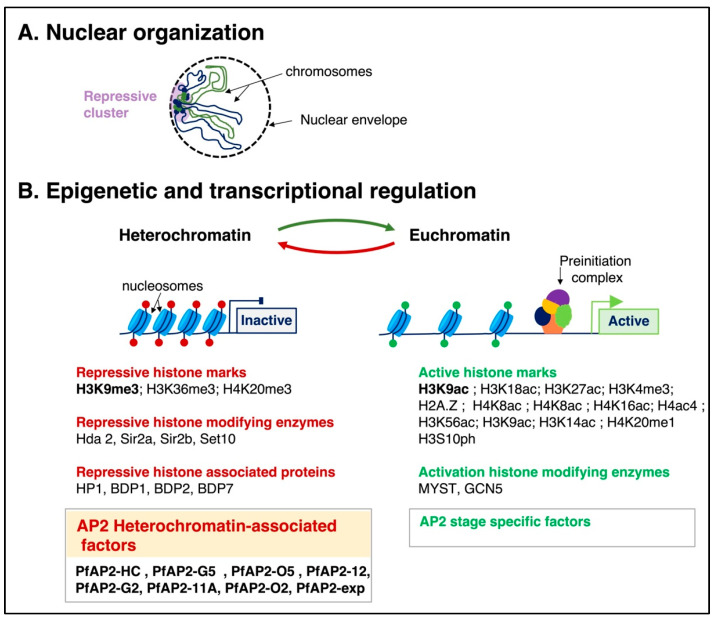
General nuclear organization, epigenetic, and transcriptional mechanisms employed by *Plasmodium* to regulate gene expression.

## Data Availability

No new data were created or analyzed in this study. Data sharing is not applicable to this article.

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
