# Peer review of "Plasmodium falciparum Development from Gametocyte to Oocyst: Insight from Functional Studies"

_microorganisms, 2023, doi:10.3390/microorganisms11081966_

Round 1

Reviewer 1 Report (New Reviewer)

The manuscript entitled "Plasmodium falciparum Development from Gametocyte to Oocyst" Title, abstract and overall rationale of work is written satisfactory. There are major concerns, which needs to be addressed before publication.

1) Abstract part: This part is too much lengthy and introductory part is more and author should be reduce upper part and write a concise way.

2) In the introduction section: Line no. 58-72 author describe about the life cycle of malaria parasite and they cite one paper and that are too old. Author try to describe the figure 1 in these line and I suggest author to incorporate the one more references in the middle and they see the A Critical Review on Human Malaria and Schistosomiasis Vaccines: Current State, Recent Advancements, and Developments paper.

3) In the figure 1 some written part is not proper align for example Microgamete, macrogamete. Male gametocyte and other. I recommend author to correct the figure. What is M?

4) Introduction part is also too much written and recommend author to reduce it and write concise way because after introduction section author divided many sub-section to explain all gene expression and gametogenesis and other.  

5) In the section 1.3 (Transcriptional regulation of gene expression) author explained details about the transcriptional regulation of the gene expression and all these written well. I request to author it is better to add one diagram here to represent all gene regulation. Moreover in other section 3.2. (Signaling cascade controlling gametogenesis) also need a mechanism figure to show clear picture and make more attraction of this review article.

6) All table font is smaller and recommend to increase the font size.

7) Conclusion part written very well

8) Some places are typographical errors I found it. kindly correct it.

9) Some references are too old and author need to revise for example reference no 5, 12, 73 and other. I suggest author to revise if other latest manuscript is available in the same information. I suggest author can be used this article related to sporozoite as a reference number 12 that is Immune responses in liver and spleen against Plasmodium yoelii pre-erythrocytic stages in Swiss mice model.

English is good written.

Author Response

We appreciate that the reviewer reviewed the manuscript and suggested revisions for improvement. We have revised the manuscript to address questions and comments as follows:

  1. Major issues:

Major issue 1 (Reviewer # 1). Abstract part: This part is too much lengthy and introductory part is more and author should be reduced upper part and write a concise way.

We agree with the reviewer, and we have reduced the introduction part of the abstract. The abstract went from 325 words to 245 words.

Major issue 2 (Reviewer # 1). In the introduction section: Line no. 58-72 author describe about the life cycle of malaria parasite, and they cite one paper and that are too old. Author try to describe the figure 1 in these line and I suggest author to incorporate the one more references in the middle and they see the A Critical Review on Human Malaria and Schistosomiasis Vaccines: Current State, Recent Advancements, and Developments paper.

Some references are too old, and author need to revise for example reference no 5, 12, 73 and other. I suggest author to revise if other latest manuscript is available in the same information. I suggest author can be used this article related to sporozoite as a reference number 12 that is Immune responses in liver and spleen against Plasmodium yoelii pre-erythrocytic stages in Swiss mice model.

In the first submitted manuscript, we cited reviews to support knowledge and statements that were accepted in the field and referenced primary literature for relatively new concepts. However, we received conflicting recommendations regarding the citations. While one of the reviewers suggested citing recent references (and dismissing all old references), a second reviewer required primary literature sources for all statements (which were often old). Although uncertain about addressing these conflicting tasks, we replaced all review citations with the primary literature (often old). When appropriate, we maintained review(s) citations to support the acceptance of the concept we refer to.

With this new submission, reviewer #1 suggested the removal of old citations.

There will always be different point of views regarding citations, and we believe that as long as the citation is correct, we should maintain them.

As suggested by reviewer#1, we have included more recent citations to cover the life cycle of the parasite in humans (PMID: 30008275) and the mosquito (PMID: 35804459, PMID: 15186403, PMID: 28108531)

Major issue 3 (Reviewer # 1). In the figure 1 some written part is not proper align for example Microgamete, macrogamete. Male gametocyte and other. I recommend author to correct the figure. What is M?.

We aligned macrogamete in the attached updated version. however, in our file, we did not see the other misaligned words nor the "M." This is likely due to the poor resolution of the first image. In the revised version, we have increased the image's resolution to 1200 dpi and hope this will have corrected the alignment issue and incomplete wording.

Major issue 4 (Reviewer # 1 ). Introduction part is also too much written and recommend author to reduce it and write concise way because after introduction section author divided many sub-sections to explain all gene expression and gametogenesis and others.

We have reduced the introduction part as follows:

  • We have shortened the section describing the life cycle.
  • We removed the paragraph introducing the concept of gene regulation, as this is developed later in the manuscript.

Major issue 5 (Reviewer #1) In the section 1.3 (Transcriptional regulation of gene expression) author explained details about the transcriptional regulation of the gene expression and all these written well. I request to author it is better to add one diagram here to represent all gene regulation. Moreover in other section 3.2. (Signaling cascade controlling gametogenesis) also need a mechanism figure to show clear picture and make more attraction of this review article.

Regarding section 1, we agree with the reviewer. We have included one diagram to represent the general gene expression regulation mechanism.

Mechanisms related to specific stage gene expressions are summarized in 4 tables. Although a graphic illustration for section 3.2 is more attractive, this will likely duplicate the information in Table 4.

Major issue 6 (Reviewer # 1). All table font is smaller and recommend to increase the font size.

We have increased the font size and resolution in the updated version of the manuscript.

  1. Minor issues:

Minor issue 1 (Reviewer # 1). Some places are typographical errors I found it. kindly correct it.

We have gone through the document and have corrected typographical errors.  

Reviewer 2 Report (New Reviewer)

This review describes the challenges of eradicating malaria, with a focus on Plasmodium falciparum. It emphasizes the need to target all stages of the parasite's life cycle, particularly transmission. The paper reviews the mechanisms involved in the development of gametocytes and their differentiation within mosquitoes. Understanding these mechanisms can help identify potential targets for developing new therapeutics. However, the full potential of epigenetics, genomics, transcriptomics, proteomics, and functional genetic studies in understanding malaria is yet to be fully realized due to limitations in using human malaria parasites and field isolates. The review can be published after addressing the following issues.

In line 75. “Female Anopheles mosquitos” should be corrected to “Female Anopheles mosquitoes

In line 81. “ultimately produce thousands of sporozoites following a process known as sporogony” should be corrected to “ultimately produce thousands of sporozoites through a process known as sporogony”

In line 90 “zygote differentiation within the mosquito will likely reveal new avenues to interrupt the parasite life cycle, thus, malaria transmission” should be corrected to "and zygote differentiation within the mosquito will likely reveal new avenues to interrupt the parasite life cycle, thus interrupting malaria transmission.”

This review describes the challenges of eradicating malaria, with a focus on Plasmodium falciparum. It emphasizes the need to target all stages of the parasite's life cycle, particularly transmission. The paper reviews the mechanisms involved in the development of gametocytes and their differentiation within mosquitoes. Understanding these mechanisms can help identify potential targets for developing new therapeutics. However, the full potential of epigenetics, genomics, transcriptomics, proteomics, and functional genetic studies in understanding malaria is yet to be fully realized due to limitations in using human malaria parasites and field isolates. The review can be published after addressing the following issues.

In line 75. “Female Anopheles mosquitos” should be corrected to “Female Anopheles mosquitoes

In line 81. “ultimately produce thousands of sporozoites following a process known as sporogony” should be corrected to “ultimately produce thousands of sporozoites through a process known as sporogony”

In line 90 “zygote differentiation within the mosquito will likely reveal new avenues to interrupt the parasite life cycle, thus, malaria transmission” should be corrected to "and zygote differentiation within the mosquito will likely reveal new avenues to interrupt the parasite life cycle, thus interrupting malaria transmission.”

Author Response

We appreciate that the reviewers reviewed the manuscript and suggested revisions for improvement. We have revised the manuscript to address questions and comments as follows:

Minor issues:

 Minor issue 1 (Reviewer # 2). In line 75. “Female Anopheles mosquitos” should be corrected to “Female Anopheles mosquitoes”

We have corrected the wording.

Minor issue 2 (Reviewer # 2). In line 81. “ultimately produce thousands of sporozoites following a process known as sporogony” should be corrected to “ultimately produce thousands of sporozoites through a process known as sporogony”.

We have corrected the wording.

Minor issue 3 (Reviewer # 2). In line 90 “zygote differentiation within the mosquito will likely reveal new avenues to interrupt the parasite life cycle, thus, malaria transmission” should be corrected to "and zygote differentiation within the mosquito will likely reveal new avenues to interrupt the parasite life cycle, thus interrupting malaria transmission.”.

We have corrected the wording.

Reviewer 3 Report (New Reviewer)

The authors demonstrate in their manuscript the Development of Plasmodium falciparum from Gametocyte to Oocyst, I found the manuscript interesting but I have the following concerns

1- What is the contribution of the manuscript to the field

2- The authors should organize the inserted tables in a better way

3- Please improve the resolution of Figure 1

4- Please insert a table where you link stages of development to recently approved or under approval drugs and specify their mechanism of action

5- A paragraph of drugs repurposed against malaria is required with a nice table

Moderate editing of English language required

Author Response

We appreciate that the reviewers reviewed the manuscript and suggested revisions for improvement. We have revised the manuscript to address questions and comments as follows:

  1. Major issues:

Major issue 1 (Reviewer # 3). What is the contribution of the manuscript to the field.

The proposed manuscript's concept is part of ongoing research by our group, which focuses on the transmission interface between humans and mosquitoes. Gametocytogenesis and Plasmodium development in the mosquito has been the subject of several comprehensive reviews. However, these reviews often focus either on gametocytogenesis in the human host or parasite development within the mosquito vector. In this review, we aim to cover Plasmodium falciparum developmental process from the pre-sexual stage in the human host to the ookinete stage of the mosquito vector. The review intended to provide a background for field-isolates research and the orientation to consider when questioning these original materials.

Major issue 2 (Reviewer # 3 ). Please insert a table where you link stages of development to recently approved or under approval drugs and specify their mechanism of action. A paragraph of drugs repurposed against malaria is required with a nice table.

While drugs repurposing is also of great interest when referring to transmission-blocking intervention, we believe that a dedicated paragraph and illustration on approved drugs or drugs under development will only lengthen the manuscript and diverge from the paper's primary objective, which is to provide insight into mechanisms used by the parasites to differentiate. Several reviews focusing on drug repurposing against malaria are available (PMID: 32731386; PMID: 33485067; PMID: 33921170).

2. Minor issues:

Minor issue 1 (Reviewer # 3). Please improve the resolution of Figure 1.

We have increased the image's resolution to 1200 dpi

Minor issue 2 (Reviewer # 3). The authors should organize the inserted tables in a better way.

We have reorganized table insertion.

Round 2

Reviewer 1 Report (New Reviewer)

The authors have addressed all the concerns raised in the previous version of the manuscript and the quality has much improved after incorporating required modifications. Therefore, the manuscript may be considered for publication in this Journal.

English is good.

Reviewer 3 Report (New Reviewer)

I am pleased to accept the manuscript in its present form

Minor editing of English language required

This manuscript is a resubmission of an earlier submission. The following is a list of the peer review reports and author responses from that submission.

Round 1

Reviewer 1 Report

This review is well-written and addressing an interesting topic. However, to make it clear for readers, adding a table or a figure summarizes the genes/proteins involved in each stage of development is important. 

Author Response

Attached, please find a revised version of the manuscript microorganisms-1945091  entitled “Plasmodium falciparum development from gametocyte to oocyst: insight from functional genetic studies”, which we wish to submit for publication in the MDPI journal microorganisms, section parasitology and the special issue Cellular Biology of Protozoan Parasites of Mammals.

We appreciate that the editors and reviewers found this work interesting and relevant to the journal. We have revised the manuscript to address their questions and comment as follow:

Minor point 1. Adding a table or a figure that summarizes the genes/proteins involved in each stage of development is important. It would be beneficial to the reader to have a figure depicting the lifecycle

We agree with the reviewer and have added a summary table and a figure as suggested.

Author Response

Attached, please find a revised version of the manuscript microorganisms-1945091  entitled “Plasmodium falciparum development from gametocyte to oocyst: insight from functional genetic studies”, which we wish to submit for publication in the MDPI journal microorganisms, section parasitology and the special issue Cellular Biology of Protozoan Parasites of Mammals.

We appreciate that the editors and reviewers found this work interesting and relevant to the journal. We have revised the manuscript to address their questions and comment as follow:

Minor point 1 (Reviewer #2). The reference list is very extensive (137) and contains a lot of old references. Are there any more recent papers to be included?

We have included additional new references whenever possible as suggested by the reviewer.

Minor point 2 . A brief sentence introducing the different strains of Plasmodium spp is recommended, especially the species that infect humans, and also mention the rodent strain P. berghei as this is mentioned several times in the review

We thank both reviewers and have included sentences to describe Plasmodium species of medical importance and those used in the laboratory for research purposes.

Minor point 3. The font size changes throughout the document, although this could be a formatting error when viewed on my mac, please check the font size

As requested by the reviewer, we have reviewed the whole manuscript and addressed the police font issue.

Minor point 4. A clearer picture of the knowledge gaps that remain, or unanswered questions in this field or directions for further study should be more clearly stated

We thank the reviewer for this important suggestion and have included a section highlighting unanswered questions and gaps in the field.

Minor point 5. Minor typos to be fixed in lines 193,203, 213, 245, 248, 256, 282, 384, 390 and 435

as All typos were corrected accordingly as follows:

  • Line 193: “…mechanism…” was corrected to “…mechanisms…”
  • Line 202: “While falciparum genome…”correct to “While the P. falciparum genome…”
  • Line 203: “…during the asexual developmental cycle…” was replaced by “during the erythrocyte cycle”
  • Line 231: “…differentiation of male gametocyte” was replaced by “…differentiation of male gametocytes…”
  • Line 182:“… The development of male and female gametocyte …” corrected to “….The development of male and female gametocytes …”
  • Line 233:“…the differentiation of female gametocyte…” corrected to “….the differentiation of female gametocytes …”
  • Line 241:“…In female gametocyte…” corrected to “….In female gametocytes …”
  • Line 241: “Plasmodium genome” corrected to “The Plasmodium genome”
  • Line 359: “Similar to eukaryotic model system…” corrected to “Similar to eukaryotic model systems…”
  • Line 365:”Ookinetes maturation” corrected to “Ookinete maturation”

Line 406: “…carried out using parasite that…” corrected to “…carried out using parasites that..”

Reviewer 3 Report

Summary: The review by Ouologuem and colleagues sets out to outline the changes and mechanisms underlying gene expression involved in Plasmodium parasite development across the sexual stages within the human host  as well as within the mosquito vector. Several comprehensive reviews already describe the mechanisms involved in regulating gene expression during the blood stages of the life cycle (including both asexual and sexual stages), however there is currently no review that covers these stages and the ones in the mosquito. The authors first highlight the mechanisms of gene regulation utilised (genome organisation, epigenetic regulation and transcriptional regulation) and then go into how these mechanisms are utilised by the different life cycle stages (gametocytes, gametes, ookinetes and oocysts). Unfortunately this work falls very short of what is expected from a review. Multiple times and in several sections the authors either cite the wrong literature or make false statements. This casts significant doubt on the accuracy and reliability of the review, especially considering this happens consistently throughout the paper. Throughout this work, the authors tend to cite reviews rather than primary sources of literature making it exceedingly difficult to determine where they have obtained their information from and establish its reliability. Additionally the authors gloss over the mechanisms underlying gene expression, failing to go into sufficient detail for this review to be of use to a reader intending to understand how Plasmodium parasites modulate gene expression. There is also a substantial amount of literature and work pertaining to gene regulation that is not mentioned - as a result the review is not a comprehensive one. The review also lack a substantial amount of background information, making this work difficult to follow for a non-expert in the field. There is also a lack of clarity in what species certain data is obtained from (be it P. falciparum or P. berghei) which is quite important to know. As a result, the authors make sweeping generalisations which are misleading or simply inaccurate.      Major Issues: The review cites the wrong literature several times and make false statement. I have tried to highlight this where possible, but this casts significant doubt as to the accuracy of this review.    In several instances the authors cite reviews rather than primary literature. Considering this is a review, this is simply unacceptable and should be amended.    The review is lacking a substantial quantity of background that would be required fro a non-expert reader to understand the paper. For example, the authors never mention what malaria actually is, who it affects, which areas it is present in. They also should explain the full life cycle before focusing only on the sexual blood and mosquito stages.    The authors mention several Plasmodium species, focusing namely on Plasmodium falciparum and Plasmodium bergheibut never addresses the fact that these species infect different hosts. Considering the fact that they infect humans and mice, respectively as well as the fact that P. falciparum is quite distinct from all other human infecting species, it is important that the authors highlight this. It is also important the findings of research papers that they highlight whether this work was done in P. falciparum or P. berghei considering that they are very different parasites.    It is unclear why the authors limit their review to the stages from gametocytes to ookinetes and do not explore sporozoites when these could in theory also be targets of transmission blocking strategies? It is also unclear why they do not discuss the asexual stages considering this is where the majority of the work has taken place and is the most informative in terms of gene regulation.    The authors highlight at some point that we need to target the mosquito stages of malaria infection in order to achieve malaria elimination. While this is true, this is conventionally achieved using bed nets and insecticides, however they argue that understanding the mechanisms of gene regulation during the mosquito stages is also required to achieve this. Can they explain how? I find this difficult to image how they would go about achieving this.     Specific Issues  Line 25 - distinct in what way? From each other or compared to the other human infecting species?    Line 31 - should the authors consider mentioning some of the statistics about malaria, highlighting how big an issue it remains? They also do not explain what malaria is or the fact that is is caused by a parasite that replicates in red blood cells. There is a lot of fundamental information about malaria that is missing and is required to put this review into perspective for the reader.    Line 33 - is reference 3 appropriate here?   Line 40 - the authors mention extreme alteration of the parasite morphology during gametocyte formation, however this is only really the case for P. falciparum gametocytes and not the other human infecting species. They should remove this statement here since they are talking generally about Plasmodium sp. and only use it when they speak specifically about Plasmodium falciparum gametocytes.    Line 44 - the way that the sentence is worded makes it seem like the sex of the parasites is only established once gametocytes convert to gametes. However, this is already determine in gametocytes. The authors should consider changing the wording of this sentence.    Lines 44-50 - Please include the appropriate references for the statements made in these lines.    Line 53 - some of the references here are not appropriate considering the authors are discussing only the mosquito and sexual blood stages. Ref 12 & 14 is about asexual parasites, Ref 13 is about transcriptional changes to drug treatment and is therefor not an appropriate reference.    Line 67 - the authors should explain what P. falciparum is - the fact that it is the most deadly human infecting species and differs significantly from other human infecting species. This should be mentioned either here or somewhere earlier in the introduction, specially considering they are making a mention of phenotypic differences between the different Plasmodium species.  Line 91 - the authors should state that P. falciparum infects humans and P. berghei infects mice and highlight why they are focusing on P. berghei studies (a lot of transmission studies done using Pb)    Line 72-74 - this sentence is a bold claim that is not necessarily substantiated by the references cited here. Firstly, their claim suggests that the entirety of the lifecycle is coordinated by changes in gene expression, when in fact the cited literature only looks at 3 life cycle stages(asexuals, gametocytes and ookinetes). Secondly, while Ref 31 does shows that there are transcriptional changes across the 3 life cycle stages they looked at, the claim that changes in gene expression is the predominant mechanism of mediating parasite development is an overstatement. The authors should lessen their claim and refrain from making and generalised statements.    Line 99 - Ref 27 is not appropriate here, there is no mention of chromatin packaging or histone modifications regulating gene expression.    Line 103-104 - I don't see any mention of the AT-rich repeat flanking the nucleosomes in Ref 35. Is it appropriate here?   Line 132 - remove putative in this sentence since it seems like only the histone acetyl-transferases are putative when in fact the rest are too.    Line 137 - considering this is a review that focuses on gene expression I find it odd that the reviewers only gloss over how this is actually achieved. In this sentence for example the authors list genes and do not explain how they achieve the silencing of gene families or cite the relevant primary literature. The authors should go into the mechanisms and at the very least cite the relevant works rather than reviews. Moreover Ref 45 and Ref46 pertain to PfSET10 and PfMYST, respectively which are not even mentioned in this sentence. To make matters worse there are no references cited that relate to Hda2 or PfSet2 which is simply unacceptable.    Line 139 - the authors should explain here what var genes are since it is the first mention of them and the reader may not know what they are. It is also particularly important to highlight the importance of var genes in P. falciparum   Line 139-142 - Ref 48 is wrong. This work related to RNA polymerase, and there is no mention of BDPs or HP1 in this article. Ref 47 and 37 are also not primary literature pertaining to HP1 or the BDPs. Therefore none of the references even mention the BDPs and it is unclear where they make the statement that the BDPs and HP1 are involved in regulating genes involved in RBC invasion and gametocyte differentiation. The fact that the authors do not even reference the BDP1/2 paper (https://www.cell.com/cell-host-microbe/pdfExtended/S1931-3128(15)00213-9) suggests that this is a complete oversight and not just a mix up in citations.   Line 139-142 & Line 145-150 - Ref 47 is a review pertaining to HP1 and not primary literature please cite the primary literature. The authors should also go into more detail as to how HP1 is involved in regulating gene expression - namely the fact that is binds to trimethylated H3K9 associated with clonally variant gene families and genes localised at the nuclear periphery, aggregating nucleosome to form heterochromatin and allowing for mutually exclusive expression of var genes (https://pubmed.ncbi.nlm.nih.gov/19270070/https://www.sciencedirect.com/science/article/pii/S1931312809000304, https://pubmed.ncbi.nlm.nih.gov/19730695/ ) (https://pubmed.ncbi.nlm.nih.gov/19270070/ )   Line 148 - also important to mention here what rif and stevor genes are.    Line 164 - what are GTFs? General Transcription Factors? Please explain abbreviations in the first instance of their use.    Line 165 - first use of TFIID please explain what it means.    Line 195-198 - stage I-V is specifically for P. falciparum - but the authors here are generally talking about Plasmodium species - this is not correct. Additionally the dramatic alterations in parasite size and shape are specific ti P. falciparum. Therefore please highlight the differences in P. falciparum gametocytes compared to other human and rodent infecting gametocytes.    Line 206 - the authors are missing some key information here namely that gametocytogenesis can be triggered metabolically since P. falciparum gametocyte formation is repressed by lysophosphatidyl choline (https://www.ncbi.nlm.nih.gov/pmc/articles/PMC5733390/). The authors are also missing key information of AP2-O3 which has been shown to be involved in regulating sex specific gene expression in female gametocytes (https://www.embopress.org/doi/full/10.15252/embr.202051660 )   Line 208-211 - this has since been disproven (https://pubmed.ncbi.nlm.nih.gov/30478286/) - gametocytes can form within the same replicative cycle.    Line 226 - why is there no mention of GDV1 in this section? This is a key molecule involved in the cascade of gametocytogenesis and should be included here.    Line 247 - like females males also need to emerge from the RBC - please mention this here.    Line 279 - it is the cGMP signalling pathway, not the guanylyl cyclase pathway.    Line 290-295 - the pathway described here (formation of IP3 and DAG) has not been studied in gamete conversion, only in asexual blood stages and in Toxoplasma. Moreover ALL the references cite here (Ref 82, 86-88) have nothing to do with the formation of IP3 and DAG following PKG activation. The citations are related to phosphorylation cascades in gametes. It is really not okay to make claims when these experiments have never been done, not only is this misleading but the authors didn't even mention the correct citations relating to IP3 and DAG.    Line 296-299 - References 80-82 have nothing to do with calcium release or calcium signalling.    Line 300 - missing citation.    Line 302 - once again references have nothing to do with CDPKs - where are these statements coming from?    Line 308 - this statement is incorrect - knockdown in CDPK1 suggests it is involved in host cell lysis not exflagellation - the authors also fail to state that CDPK1 translationally regulates mRNAs in the developing zygote.    Line 328 - do the authors mean NEK1 and NEK3 rather than NEK2 and 4? They only make mention of NEK1 and 3 earlier not 2 and 4.    Line 345 - REf 114 is incorrect. Please be more careful when citing papers - there is no mention of P48/45 or hap2/gcs1 in this citation.    Line 349 - this is the first mention of IMC and it is not explained adequately. How is the reader supposed to know what an IMC is or what it does? Please explain this    Line 355 - this is the wrong citation again.    Line 365 - this is the wrong citation for PK7 - it should be https://journals.asm.org/doi/abs/10.1128/ec.00245-07   Line 402-403 - how exactly do the authors suggest that we target ookinetes? please clarify    Line 407-410 - citation?         

Author Response

Attached, please find a revised version of the manuscript microorganisms-1945091  entitled “Plasmodium falciparum development from gametocyte to oocyst: insight from functional genetic studies”, which we wish to submit for publication in the MDPI journal microorganisms, section parasitology and the special issue Cellular Biology of Protozoan Parasites of Mammals.

We have revised the manuscript to the reviewer's major and minor questions as follows:

  1. Major issues:

Major issue 1. The review cites the wrong literature several times and make false statement. I have tried to highlight this where possible, but this casts significant doubt as to the accuracy of this review.    In several instances the authors cite reviews rather than primary literature. Considering this is a review, this is simply unacceptable and should be amended.   

We thank the reviewer for her (his) thoughtful reading of the manuscript and have updated the document appropriately. All the original source papers were updated and cited accordingly.

Major issue 2. The review is lacking a substantial quantity of background that would be required for a non-expert reader to understand the paper

The original intent of the review was to target readers with a background on malaria parasite(s) that infect humans and others used in mouse studies. However, we have addressed the issues by providing a description and a figure of the parasite life cycle.

Major issue 3. It is also important the findings of research papers that they highlight whether this work was done in P. falciparum or P. berghei considering that they are very different parasites

We agree with the reviewer and have thoroughly reviewed the manuscript by highlighting the malaria species when necessary.

 Major issue 4. It is unclear why the authors limit their review to the stages from gametocytes to ookinetes and do not explore sporozoites when these could in theory also be targets of transmission-blocking strategies?

We agree with the reviewer that exploring the sporozoite stage of the parasite could have been more exhaustive. However, as stated in the introduction, we set the goal of focusing the review on malaria parasite stages that we were interested into.

Major issue 5. It is also unclear why they do not discuss the asexual stages considering this is where the majority of the work has taken place and is the most informative in terms of gene regulation?

As stated above, we aim to focus on stages of interest for our ongoing studies. In addition, they are extensive reviews on malaria asexual stages, and this review aims to highlight gaps in coverage.

Major issue 6. The authors highlight at some point that we need to target the mosquito stages of malaria infection in order to achieve malaria elimination. While this is true, this is conventionally achieved using bed nets and insecticides, however they argue that understanding the mechanisms of gene regulation during the mosquito stages is also required to achieve this. Can they explain how? I find this difficult to image how they would go about achieving this?

We agree with the reviewer and have updated the whole paragraph to show that the control of the mosquito stage can be accomplished using mostly conventional techniques.

2. Minor issues:

Minor point 1 . Adding a table or a figure summarizes the genes/proteins involved in each stage of development is important. It would be beneficial to the reader to have a figure depicting the lifecycle

We agree with the reviewer and have added a summary table and a figure as suggested.

Minor point 2. A brief sentence introducing the different strains of Plasmodium spp is recommended, especially the species that infect humans, and also mention the rodent strain P. berghei as this is mentioned several times in the review

We thank both reviewers and have included sentences to describe Plasmodium species of medical importance and those used in the laboratory for research purposes.

Minor point 3. The font size changes throughout the document, although this could be a formatting error when viewed on my mac, please check the font size

As mentioned earlier, we have gone through the document and addressed the police font issue.

Minor point 4. Line 31 – Should the authors consider mentioning some of the statistics about malaria, highlighting how big an issue it remains? They also do not explain what malaria is or the fact that is caused by a parasite that replicates in red blood cells. There is a lot of fundamental information about malaria that is missing and is required to put this review into perspective for the reader.

We agree with the reviewer and have provided some statistics about malaria. We have defined malaria and provided additional information to put the review into perspective for the reader.

Minor point 5. Line 33 – is reference 3 appropriate here

We agree with the reviewer and have removed the reference.

Minor point 6. Line 40 – the authors mention extreme alteration of the parasite morphology during gametocyte formation; however, this is only really the case for P. falciparum gametocytes and not the other human infecting species. They should remove this statement here since they are talking generally about Plasmodium sp. and only use it when they speak specifically about Plasmodium falciparum gametocytes.    

We agree with the reviewer and have clarified the statement.

Minor point 7. Line 44 - the way that the sentence is worded makes it seem like the sex of the parasites is only established once gametocytes convert to gametes. However, this is already determined in gametocytes. The authors should consider changing the wording of this sentence

We agree with the reviewer and have make the sentence clearer.

Minor point 8. Lines 44-50 - Please include the appropriate references for the statements made in these lines.  

We agree with the reviewer, and we have included appropriate references for the sentences.

Minor point 9. Line 53 - some of the references here are not appropriate considering the authors are discussing only the mosquito and sexual blood stages. Ref 12 & 14 is about asexual parasites, Ref 13 is about transcriptional changes to drug treatment and is therefore not an appropriate reference

We agree with the reviewer, and we have included appropriate references for the sentences.

Minor point 10. Line 67 - the authors should explain what P. falciparum is - the fact that it is the deadliest human infecting species and differs significantly from other human infecting species. This should be mentioned either here or somewhere earlier in the introduction, especially considering they are making a mention of phenotypic differences between the different Plasmodium species.

We have included sentences to describe Plasmodium species of medical importance in the introduction.

Minor point  11. Line 91 - the authors should state that P. falciparum infects humans and P. berghei infects mice and highlight why they are focusing on P. berghei studies (a lot of transmission studies done using Pb)

We have included sentences highlighting the rodent Plasmodium berghei and its use as a malaria model system.

Minor point 12. Line 72-74 - this sentence is a bold claim that is not necessarily substantiated by the references cited here. Firstly, their claim suggests that the entirety of the lifecycle is coordinated by changes in gene expression, when in fact the cited literature only looks at 3 life cycle stages (asexuals, gametocytes and ookinetes). Secondly, while Ref 31 does shows that there are transcriptional changes across the 3 life cycle stages they looked at, the claim that changes in gene expression is the predominant mechanism of mediating parasite development is an overstatement. The authors should lessen their claim and refrain from making and generalized statements.   

We have provided new references to support our claims.

Minor point 13 Line 99 - Ref 27 is not appropriate here, there is no mention of chromatin packaging or histone modifications regulating gene expression.

We agree with the reviewer, and we have included appropriate references for the statement.

Minor point 14. Line 103-104 - I don't see any mention of the AT-rich repeat flanking the nucleosomes in Ref 35. Is it appropriate here?    

We agree with the reviewer, and we have removed the reference.

Minor point 15. Line 132 - remove putative in this sentence since it seems like only the histone acetyl-transferases are putative when in fact the rest are too.

We agree with the reviewer, and we have removed “putative”.

Minor point 16. Line 137 - considering this is a review that focuses on gene expression I find it odd that the reviewers only gloss over how this is actually achieved. In this sentence for example the authors list genes and do not explain how they achieve the silencing of gene families or cite the relevant primary literature. The authors should go into the mechanisms and at the very least cite the relevant works rather than reviews. Moreover Ref 45 and Ref46 pertain to PfSET10 and PfMYST, respectively which are not even mentioned in this sentence. To make matters worse there are no references cited that relate to Hda2 or PfSet2 which is simply unacceptable

We agree with the reviewer that providing a detailed mechanism on how gene silencing is achieved would have been necessary if the focus of the review paper was on gene expression. However, a detailed explanation for all the players might diverge the review's focus in this introductory paragraph. The primary goal was the goal of this review is to highlight gaps in coverage. Therefore, we have provided appropriate references for each player for the reader to refer to for more explanation.

Minor point 17. Line 139 - the authors should explain here what var genes are since it is the first mention of them and the reader may not know what they are. It is also particularly important to highlight the importance of var genes in P. falciparum    

We agree with the reviewer, and we have provided an explanation for  var genes.

Minor point 18 Line 139-142 - Ref 48 is wrong. This work related to RNA polymerase, and there is no mention of BDPs or HP1 in this article. Ref 47 and 37 are also not primary literature pertaining to HP1 or the BDPs. Therefore none of the references even mention the BDPs and it is unclear where they make the statement that the BDPs and HP1 are involved in regulating genes involved in RBC invasion and gametocyte differentiation. The fact that the authors do not even reference the BDP1/2 paper (https://www.cell.com/cell-host-microbe/pdfExtended/S1931-3128(15)00213-9) suggests that this is a complete oversight and not just a mix up in citations.  

We agree with the reviewer. We have updated the paragraph and provided primary literature for the different protein described.

Minor point 19. Line 139-142 & Line 145-150 - Ref 47 is a review pertaining to HP1 and not primary literature please cite the primary literature. The authors should also go into more detail as to how HP1 is involved in regulating gene expression - namely the fact that is binds to trimethylated H3K9 associated with clonally variant gene families and genes localized at the nuclear periphery, aggregating nucleosome to form heterochromatin and allowing for mutually exclusive expression of var genes:

https://pubmed.ncbi.nlm.nih.gov/19270070/https://www.sciencedirect.com/science/article/pii/S1931312809000304,

https://pubmed.ncbi.nlm.nih.gov/19730695/ ) (https://pubmed.ncbi.nlm.nih.gov/19270070/ )  

We have updated the paragraph and provided primary literature for the different protein described. We agree with the reviewer that providing a detailed mechanism on how HP1 works would have been necessary if the focus of the review paper was on var gene expression. However, a detailed explanation might diverge the paragraph toward var genes which is not the focus of the paragraph.

Minor point 20. Line 164 - what are GTFs? General Transcription Factors? Please explain abbreviations in the first instance of their use e.

We agree with the reviewer, and we have provided an explanation for the abbreviation.

Minor point 24 (Reviewer #3). Line 165 - first use of TFIID please explain what it means

We agree with the reviewer, and we have provided an explanation for the abbreviation.

Minor point 21. Line 195-198 - stage I-V is specifically for P. falciparum - but the authors here are generally talking about Plasmodium species - this is not correct. Additionally the dramatic alterations in parasite size and shape are specific ti P. falciparum. Therefore please highlight the differences in P. falciparum gametocytes compared to other human and rodent infecting gametocytes.     

We agree with the reviewer, and we have clarified the statement

Minor point 22. Line 206 - the authors are missing some key information here namely that gametocytogenesis can be triggered metabolically since P. falciparum gametocyte formation is repressed by lysophosphatidyl choline

 (https://www.ncbi.nlm.nih.gov/pmc/articles/PMC5733390/).

The authors are also missing key information of AP2-O3 which has been shown to be involved in regulating sex specific gene expression in female gametocytes

(https://www.embopress.org/doi/full/10.15252/embr.202051660 )

We agree with the reviewer. We have provided additional information with appropriate references.

Minor point 23. Line 208-211 - this has since been disproven (https://pubmed.ncbi.nlm.nih.gov/30478286/) - gametocytes can form within the same replicative cycle.

We agree with the reviewer. We have updated the paragraph and provided references.

Minor point 24. Line 226 - why is there no mention of GDV1 in this section? This is a key molecule involved in the cascade of gametocytogenesis and should be included here.    

We agree with the reviewer. We have updated the paragraph and provided references.

Minor point 25. Line 247 - like females males also need to emerge from the RBC - please mention this here.  OK   

We agree with the reviewer. We have updated the paragraph and provided references.

Minor point 26. Line 279 - it is the cGMP signaling pathway, not the guanylyl cyclase pathway.   

We agree with the reviewer. We have corrected the wording.

Minor point 27. Line 290-295 - the pathway described here (formation of IP3 and DAG) has not been studied in gamete conversion, only in asexual blood stages and in Toxoplasma. Moreover ALL the references cite here (Ref 82, 86-88) have nothing to do with the formation of IP3 and DAG following PKG activation. The citations are related to phosphorylation cascades in gametes. It is really not okay to make claims when these experiments have never been done, not only is this misleading but the authors didn't even mention the correct citations relating to IP3 and DAG

We are providing additional references to support the formation of IP3 and DAG following PKG activation.

Minor point 28 Line 296-299 - References 80-82 have nothing to do with calcium release or calcium signalling.  

We agree with the reviewer. We have updated the citations.

Minor point  29. Line 300 - missing citation  

We agree with the reviewer. We have provided the references.

Minor point 30. Line 302 - once again references have nothing to do with CDPKs - where are these statements coming from?

We agree with the reviewer. We have provided the references.

Minor point 31 Line 308 - this statement is incorrect - knockdown in CDPK1 suggests it is involved in host cell lysis not exflagellation - the authors also fail to state that CDPK1 translationally regulates mRNAs in the developing zygote

We agree with the reviewer. We have clarified the statement and provided references.

Minor point 32. Line 328 - do the authors mean NEK1 and NEK3 rather than NEK2 and 4? They only make mention of NEK1 and 3 earlier not 2 and 4

We have clarified the statement and provided references.

Minor point 33. Line 345 - REf 114 is incorrect. Please be more careful when citing papers - there is no mention of P48/45 or hap2/gcs1 in this citation

We agree with the reviewer. We have provided the correct references.

Minor point 34. Line 349 - this is the first mention of IMC and it is not explained adequately. How is the reader supposed to know what an IMC is or what it does? Please explain this

We agree with the reviewer. We have explained the term.

Minor point 35. Line 355 - this is the wrong citation again.

We agree with the reviewer. We have provided the correct references.

Minor point 36. Line 365 - this is the wrong citation for PK7 - it should be https://journals.asm.org/doi/abs/10.1128/ec.00245-07  

We agree with the reviewer. We have provided the correct references.

Minor point 37. how exactly do the authors suggest that we target ookinetes? please clarify

We have clarified strategies to target ookinetes.

We hope that these revisions will render the attached manuscript suitable for publication

We hope that these revisions will render the attached manuscript suitable for publication. Please feel free to contact me if there is any further information that I can provide.

Sincerely

Round 2

Reviewer 3 Report

Despite significant changes that significantly improve the quality of the manuscript, the review remains deeply flawed due to the fact that many portions of the review deceive the reader in different ways, either by citing false information, by obscuring or leaving information out, or by suggesting that experiments were done in P. falciparum when in fact they were performed in P. berghei or P. yoelii. I highlighted this in the first review and instead of addressing this, the authors appear to have ignore this and even make matters worse in some instances. For example in their abstract, they suggest that this review highlights the mechanisms involved in Plasmodium falciparum stage-specific expression (lines 22-23) when in fact the majority of the works cited have been performed in other species. Another example is when the authors highlight the importance of PfAP2-O3 in programming gender-specific transcription program (lines 319-322), suggesting this research was done in P. falciparum when it was performed in P. yoelii.    In places where I highlighted that the authors had cited the wrong information from primary literature (for example the fact that CDPK1 is involved in exflagellation), they have changed their statements, but these new statements are also wrong despite the fact that I gave them the correct information (they state that CDPK1 is involved in axoneme assembly when there is no mention of this in the primary literature the cite!). Once again this casts significant doubt into the accuracy of this review. I have only hand picked some of the references to re-review and many remain wrong.    Finally, considering the fact that the aim of this review is to highlight the gaps in coverage of our understanding of how the sexual stages develop, I still find that the review does not delve into enough detail. The authors themselves state that readers can follow the works cited to find out more information for themselves. This completely defeats the point of a review, especially when the authors cite the wrong primary literature several times and misrepresent the findings.